# Small molecule binding to surface-supported single-site transition-metal reaction centres

M. DeJong [1,2], A. J. A. Price [3], E. Mårsell [2], G. Tom [1,2], G. D. Nguyen[2], E. R. Johnson [3] ✉ & S. A. Burke [1,2,4] ✉

Despite dominating industrial processes, heterogeneous catalysts remain challenging to characterize and control. This is largely attributable to the diversity of potentially active sites at the catalyst-reactant interface and the complex behaviour that can arise from interactions between active sites. Surface-supported, single-site molecular catalysts aim to bring together benefits of both heterogeneous and homogeneous catalysts, offering easy separability while exploiting molecular design of reactivity, though the presence of a surface is likely to influence reaction mechanisms. Here, we use metal-organic coordination to build reactive Fe-terpyridine sites on the Ag(111) surface and study their activity towards CO and $C_2H_4$ gaseous reactants using low-temperature ultrahigh-vacuum scanning tunnelling microscopy, scanning tunnelling spectroscopy, and atomic force microscopy supported by density-functional theory models. Using a site-by-site approach at low temperature to visualize the reaction pathway, we find that reactants bond to the Fe-tpy active sites via surface-bound intermediates, and investigate the role of the substrate in understanding and designing single-site catalysts on metallic supports.

Surfaces provide an exciting space for chemical reactions. While the separability and reuse advantages of heterogeneous catalysts are well-recognized, the inherent symmetry breaking, decreased coordination, and distinct electronic environment at surfaces promotes reactivity. However, the diverse array of sites−i.e., step edges, kinks, vacancies, adsorbates, and defects−and interactions between reactants, products, and catalyst on even single-crystal surfaces makes both understanding and controlling reactivity an atomic-scale persuit[1–4]. Scanning tunnelling microscopy (STM) has served as an important tool for understanding the connections between structure, electronic states, and reactivity in heterogeneous catalysis since its early adoption. The ability to directly image and probe the often complicated reactive landscapes of real surfaces has yielded important insights[1–17]. While a handful of groups have developed *in operando* STMs for tracking surface reactions under realistic reactor conditions[8,10,12,18,19], STM studies under conditions that differ substantially from typical reaction conditions (i.e., low temperature and ultra-high vacuum) have

nevertheless yielded important insight into understanding of surface chemistry of single-crystal surfaces[1,3]. More recently, STM has also been used to track small-molecule activation at individual metallo-porphyrins and phthalocyanines[20–25], providing important information for designing reactive sites based on these metal-organic species. Adding to the insights provided by STM, the development of non-contact atomic force microscopy (ncAFM) imaging with CO-functionalized tips over the past decade has provided a new tool for imaging bonds[26,27], identifying molecular structure[28,29], and investigating surface mediated reactions[30–34]. Collectively, these tools provide a unique local chemical view, including the ability to track individual sites, identify changes in structure (ncAFM), and visualize the frontier molecular orbitals responsible for reactivity through both STM imaging and scanning tunnelling spectroscopy (STS)[1].

STM and ncAFM have also played a substantial role in characterizing a wide range of self-assembled nanostructures, advancing the field, and leading to exquisite control over surface-bound molecular and metal-

[1]Department of Physics & Astronomy, University of British Columbia, Vancouver, BC V6T 1Z1, Canada. [2]Stewart Blusson Quantum Matter Institute, University of British Columbia, Vancouver, BC V6T 1Z4, Canada. [3]Department of Chemistry, Dalhousie University, Halifax, NS B3H 4R2, Canada. [4]Department of Chemistry, University of British Columbia, Vancouver, BC V6T 1Z1, Canada. ✉e-mail: Erin.Johnson@Dal.Ca; saburke@phas.ubc.ca

organic structures[35–37]. Such high-fidelity on-surface coordination chemistry and related metal-organic networks offer a potential route to designing reactivity similar to solution-based molecular counterparts used in homogeneous catalysis. Indeed, the emerging field of single-site heterogeneous catalysis takes advantage of molecular design concepts from homogeneous catalysts to prepare relatively well-isolated and controlled metal-organic sites for reactivity that are anchored to surfaces, typically through solution-based chemistry on oxide surfaces[38–41]. This control borrowed from homogeneous catalysis is appealing, however, even with a relatively benign surface, changes in reactivity brought about by the surface environment[41,42] or participation of surface atoms[41,43] offer up complexity alongside opportunities for reaction design if the interaction with the surface is well understood.

Among the well-explored metal-organic coordination motifs for both solution complexes and surface-bound structures, metal-polypyridyl complexes offer a highly tunable platform[44–47], with demonstrated applications ranging from energy harvesting to medical imaging and catalysis, with a variety of morphologies from single molecules to metal-organic frameworks and polymer anchored sites[45,46,48], as well as a variety of nanostructures on surfaces[49–52]. Interest in these complexes from the catalysis community include applications in photocatalysis and electrocatalysis, and for a variety of reactions including cross-couplings, epoxide ring openings and epoxidation, cycloaddition, transfer hydrogenation, and ethylene oligomerization driven by a variety of ligand and metal combinations, and using different structural motifs[41,44,45,47,48,53–55]. In particular, early transition metals with weak-field pyridine based ligands have shown exciting prospects for promoting difficult reactions such as [2 + 2] cycloadditions (Fe and Co)[56], and other C–C coupling reactions[41] including ethylene oligomerization (Fe, Co, Ni)[57], C-H activation (Fe)[54], as well as electrocatalytic reduction of $CO_2$ (Fe, Ni, Cr, Co)[44] and oxygen reduction (Fe)[58].

Here, we investigate the role of a minimally reactive noble-metal surface in the early mechanistic steps of the attachment of model reactants (CO and ethylene) to an early transition metal (iron) and terpyridine-based ligand complex to provide insight into how the presence of a surface modifies reactivity of the ligand-bound metal site.

## Results

### Preparation and characterization of active Fe-terpyridine site

The specific system studied consists of a single Fe atom coordinated through in situ on-surface preparation to a terminal terpyridine (tpy) group in a terpyridine-phenyl-terpyridine [4',4'''-(1,4- Phenylene) bis(2,2':6',2''-terpyridine), denoted TPT] molecule on a Ag(111) substrate (Fig. 1c). Prepared entirely in ultrahigh vacuum (UHV), the iron is expected to have an incomplete coordination sphere, and a nominal charge state of $2^+$ based on previous results with a similar terpyridine ligand[51], differing from that of pure Fe atoms[17] or clusters on the surface. The electronic structure and charge state of this complex are addressed further in the discussion. Active Fe-tpy sites were characterized using low-temperature, UHV STM, Scanning Tunnelling Spectroscopy (STS), and CO functionalized ncAFM imaging. Details on the preparation of active Fe-tpy sites are described in the Methods section. Figure 1 shows STM topographs (a, d), ncAFM frequency shift images (b, e), and proposed chemical structures (c, f) for coordinated TPT molecules containing one (a–c) and two (d–f) active Fe-tpy sites. Topographic STM images show differences between the bare tpy (Fig. 1a left, blue circle indicator) and Fe-coordinated tpy (Fig. 1a right, red 4-point star, and Fig. 1d, both ends). Fe-tpy has an apparent height 18 pm taller than the bare tpy at a sample bias of 20 mV measured with a CO-terminated tip, accompanied by a characteristic electronic signature discussed later. The depressions, or shadows, appearing adjacent to distal pyridines without Fe also disappear when Fe is present. These differences in STM contrast are consistent with previous results for a similar bis-tpy molecule, and are attributed to the rotation of distal pyridines to coordinate to the Fe atom and the appearance of an Fe-tpy state slightly below the Fermi energy[50,51]. The ncAFM images further resolve these differences in the tpy structure: the bare tpy shows three clearly distinct rings, where the Fe-tpy loses some internal resolution, likely due to a small geometrical relaxation towards the surface as observed in similar works on metal-terpyridine coordination complexes[51,59]. These characteristic imaging changes, particularly in topographic images, allow us to readily identify coordinated and bare tpy groups.

### Interaction of CO and $C_2H_4$ with Fe-terpyridine site

To explore reactivity, the prepared Fe-tpy sites were exposed within the low-temperature STM head to low concentrations of gaseous CO or $C_2H_4$ via a leak valve. CO was dosed either separately or together with $C_2H_4$ in different experiments to isolate the reactants and interactions between them. CO is present in all of our experiments as a low partial pressure is typically observed in UHV; to achieve CO-functionalized imaging, additional CO was also deliberately introduced to the imaging chamber. During reactant dosing, the sample temperature rose from the base temperature of 4.2 K to 7.0–11.6 K. The

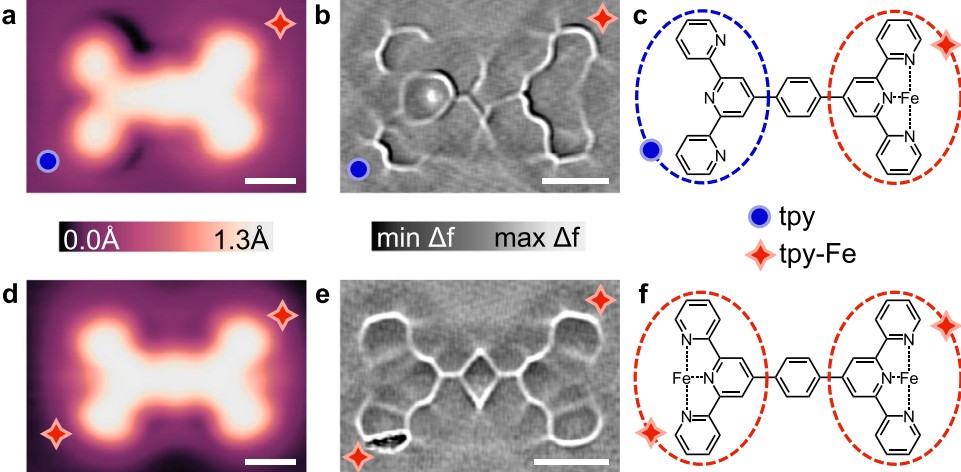

**Fig. 1 | STM and ncAFM of Fe-tpy sites.** TPT molecule with one (**a**–**c**) and two (**d**–**f**) active Fe-tpy sites. STM images (**a**, **d**) acquired with CO-functionalized tips. STM topograph with single Fe-tpy site (**a**) acquired at $V = 20$ mV and $I = 10$ pA indicates an increase in apparent height at the Fe-tpy coordinate bond (**a**, red four-point star) compared to the non-metalated tpy group (**a**, blue circle). Laplace-filtered ncAFM $\Delta f$ images ($A_{vib}$-2 Å) show reduced ring definition at the Fe-tpy sites (**b**, **e**, red 4-point star) compared with the three distinguishable pyridines of the bare tpy group (**b**, blue circle) consistent with previous work.

samples containing active sites and gaseous reactants were then annealed for 5 min intervals at temperatures incrementing in steps of $\Delta T = 5$ K from $T = 15$ K to $T = 40$ K. Between each annealing step, the sample was probed at 4.2 K, revealing changes to the physical and electronic structures of the active sites. Scalebars: 5Å (a, b, d, e).

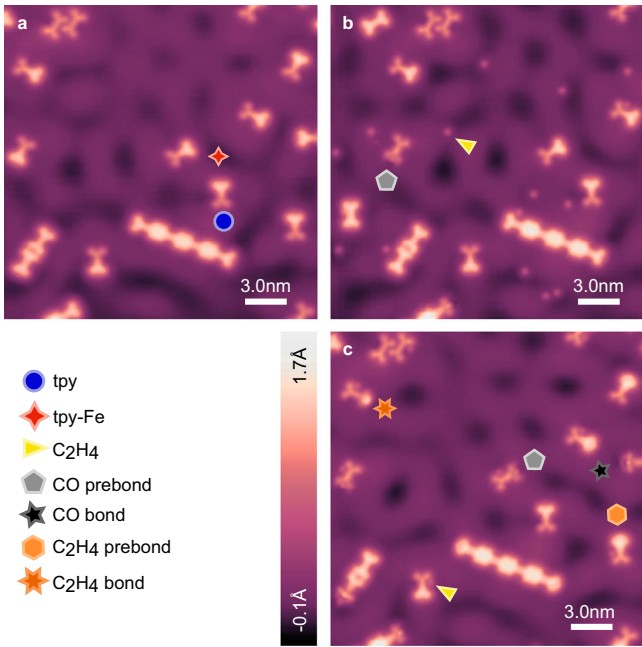

**Fig. 2 | STM overviews of reaction progression.** STM topographs of the same region of a sample imaged after active Fe-tpy site preparation (**a**), after subsequent exposure to CO and C₂H₄ (maximum temperature during dosing 11.1 K) (**b**), and after successive incremental sample annealing up to $T = 30$ K for $t = 5$ min (**c**). Free C₂H₄ (yellow triangle) is visible on the surface, along with changes to active sites labelled "CO prebond" (grey pentagon), after gas exposure (**b**). After subsequent annealing of the sample at $T = 30$ K for $t = 5$ min (**c**) three additional distinct changes to active tpy-Fe sites are visible and labelled: "CO bond" (black five point star), "C₂H₄ prebond" (light orange hexagon), and "C₂H₄ bond" (orange six point star). After the anneal, C₂H₄ molecules can form pseudo hydrogen bonds with the nitrogen lone pairs of the distal, outward pointing pyridyl of non-metalated tpy groups (**c**, yellow triangle). Images shown were acquired with a metal tip at $V = 20$ mV and $I = 20$ pA (**a–c**).

Figure 2 shows the evolution of active sites in a region of a sample exposed to both CO and C₂H₄, and subsequently annealed. The sample before dosing is shown (Fig. 2a) with an Fe-tpy site (red four-point star) and bare tpy group (blue circle) indicated for reference. Chains of TPT molecules linked with Fe nodes[50,51] are also visible in Fig. 2, but reactivity at chain centres will not be discussed in this work due to variability of the as-prepared nodes making reactant attachment difficult to distinguish. Further, indications of changes at these sites were rare, likely due to a higher diffusion barrier over the ligand for the reactants to reach the Fe. Figure 2b shows the same region of the sample from Fig. 2a after exposure to CO and C₂H₄, during which the sample temperature reached a maximum of 11.6 K. In Fig. 2b, isolated C₂H₄ (yellow triangle) molecules are visible on the surface, along with changes to some active sites, e.g., as indicated by the grey pentagon labelled "CO prebond", to be discussed in Fig. 3. Figure 2c shows the same region of the sample after incrementally annealing up to $T = 30$ K for $t = 5$ min at each step (see Supplementary Fig. 3 for schematic of full progression), where we see evidence of diffusion, and additional changes to active sites are apparent. There are seven distinct and commonly observed features in Fig. 2c, along with a few less common features. In addition to the previously identified tpy (blue circle) and Fe-tpy sites (red four-point star), there are four new features labelled: CO prebond (grey pentagon), CO bond (black five-point star), C₂H₄ prebond (light orange hexagon), and C₂H₄ bond (orange six-point star), as well as several C₂H₄ molecules near the distal pyridines of non-metallated tpy groups (yellow triangle). As the C₂H₄ near distal pyridines are easily dissociated and appear to be only slightly favoured over other sites, we will not discuss this weak interaction with the bare tpy further. The four new, most common features that appear at Fe sites after gas dosing and annealing up to $T = 30$ K will be discussed in detail in the following.

To assign each of the four distinct features labelled in Fig. 2c to CO or C₂H₄ attachment, experiments were performed where either CO only, or C₂H₄ + CO, was dosed on samples containing Fe-tpy active sites. Figure 3 shows the two most common, repeatedly observed new features on samples where only CO was dosed, denoted CO prebond (Fig. 3a–c) and CO bond (Fig. 3d–f). STM topographs (Fig. 3a, d) and ncAFM images (Fig. 3b, e) were all acquired with CO-terminated tips. The CO prebond STM topograph (Fig. 3a) shows a localized dark termination at the Fe site, visually consistent with previous findings for Fe-tpy interaction with CO[51]. Unlike for CO near isolated Fe adatoms (nominally expected to be in the Fe⁰ state) on the surface[17], these "prebond" structures interacting with Fe-tpy sites show a distinct

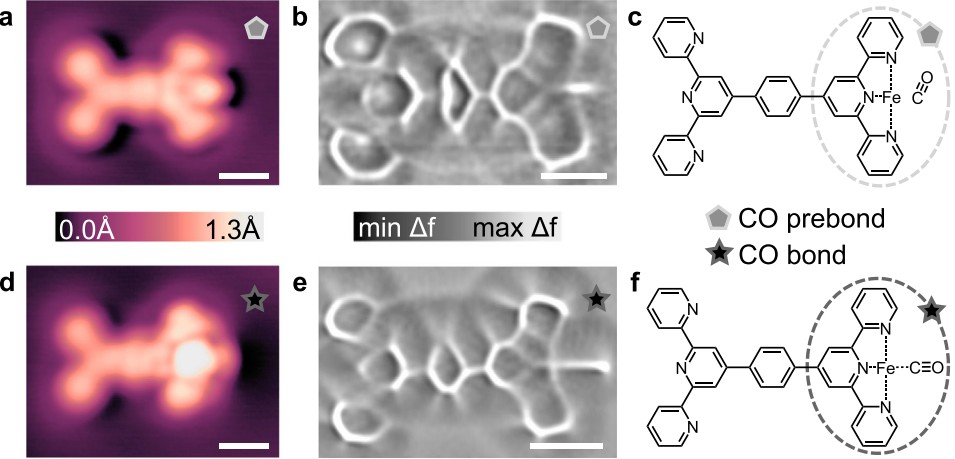

**Fig. 3 | STM and ncAFM of CO associated Fe-tpy structures.** Comparison of CO prebond (**a–c**) and CO bond (**d–f**) configurations. STM topographs (**a, d**) were imaged with CO-terminated tips at $V = 20$ mV and $I = 10$ pA. Constant height ncAFM images (Laplace-filtered) for the CO prebond (**b**) and CO bond (**e**) motifs show a distinct difference in the length of lines at the right side of the molecules where the CO is located. The proposed chemical structures for the CO prebond and CO bond configurations are shown in (**c**) and (**f**), respectively. Parameters for (**b**): $V = -0.95$ mV, $A_{vib} = 2$ Å, $\Delta z = -0.01$ nm from setpoint $V = 20$ mV and $I = 10$ pA on Ag(111). Parameters for (**e**): $V = -0.95$ mV, $A_{vib} = 8$ Å, $\Delta z = -0.06$ nm from setpoint $V = 20$ mV and $I = 10$ pA on Ag(111). Scalebars: 6.0 Å (**a, d**) 5.0 Å (**b, e**).

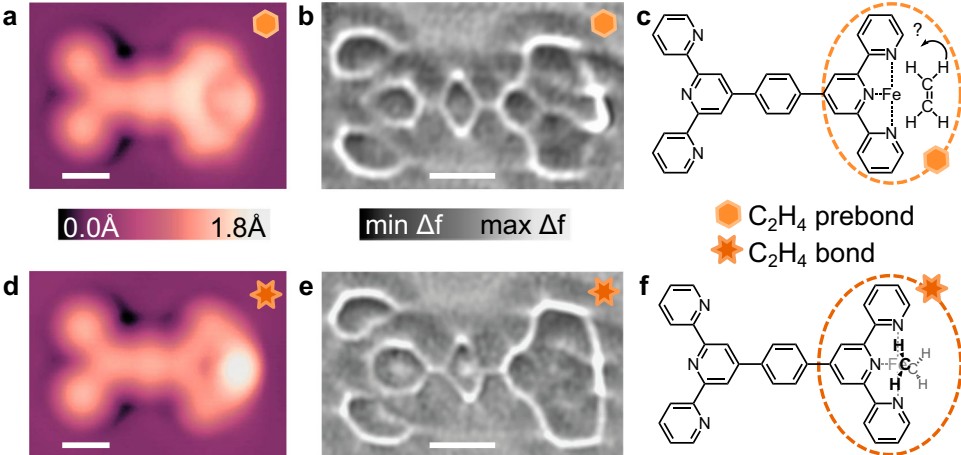

**Fig. 4 | STM and ncAFM of C₂H₄ associated Fe-tpy structures.** Topographs of the C₂H₄ prebond (**a**) and C₂H₄ bond (**d**) configurations imaged at $V = 20$ mV and $I = 10$ pA. Laplace-filtered ncAFM images of the C₂H₄ prebond (**b**) and C₂H₄ bond (**e**) motifs imaged at constant height with $A_{vib} = 2$ Å from a setpoint of $V = 20$ mV and $I = 10$ pA on Ag(111). $\Delta z = -0.01$ nm for (**b**), and $\Delta z = -0.022$ nm for (**e**). Images (**a**, **b**, **d**, **e**) were all acquired with the same CO-terminated tip. Shear transformation performed on (**b**) to compensate for drift. Proposed chemical structures for the horizontal C₂H₄ prebond and vertical C₂H₄ bond motifs drawn in (**c**) and (**f**), respectively; note for the pre-bond structure, surface adsorption would indicate a flat-lying structure as shown, while DFT shows a weakly bonded side-orientation. Scalebars 5 Å (**a**, **b**, **d**, **e**).

change in image character with a significant decrease in apparent height over the Fe (compare for example Fig. 2 panels a and b left side where the CO pre-bond is formed in panel b denoted by the grey pentagon, and again from b to c on the right side). The CO bond STM topograph (Fig. 3d) is again distinct, showing a brighter, half-ring shaped termination. The ncAFM image of the CO bond (Fig. 3e) has a distinctly longer feature measuring 4.5 Å, in comparison to the short 2.6 Å line visible at the CO prebond site in (Fig. 3b). Assuming an initial "standing up" configuration where the CO remains predominantly bonded to the surface Ag atoms (Fig. 3c), the increased line length is consistent with the expected conformational change upon CO bonding with Fe-tpy (Fig. 3f). Detailed DFT modelling, discussed later, confirms these two structures. As the proportion of CO bond species increases relative to the CO prebond upon successive annealing steps above $T = 30$ K, we attribute the prebond structure to a metastable surface-bound intermediate, with a configuration similar to the expected surface adsorption, en route to the Fe-bonded structure shown in Fig. 3f.

Experiments where C₂H₄ was dosed on samples containing active Fe-tpy sites, along with small quantities of CO as noted previously, yielded four common structures. In addition to the CO prebond and bond motifs identified in Fig. 3, two new configurations, similarly denoted C₂H₄ prebond and C₂H₄ bond, were consistently apparent after dosing and annealing. A comparison of these two structures is shown in Fig. 4 with the C₂H₄ prebond in (Fig. 4a–c) and the C₂H₄ bond in (Fig. 4d–f). Both STM topographs (Fig. 4a, d) and ncAFM images (Fig. 4b, e) were acquired with the same CO-terminated tip. The C₂H₄ prebond STM topograph (Fig. 4a) shows a rounded termination of uniform intensity, whereas the C₂H₄ bond STM topograph (Fig. 4d) has a rounded termination with a localized bright region measuring an apparent height of 0.46 Å (at 20 mV sample bias) above the long molecular axis. The ncAFM images show a more outward termination for the C₂H₄ prebond (Fig. 4b) with a "handle-like" shape compared with the C₂H₄ bond (Fig. 4e) which is nearly linear. Annealing to at least 20 K was required to mobilize C₂H₄ on the surface, and both structures only appear in significant numbers after annealing to 30 K, with a growing population of the C₂H₄ bond appearing after increased annealing to 35 K and above. The expected adsorption of C₂H₄ on Ag(111) is flat, with the π-system parallel to the surface[60], while gas-phase calculations would place the C₂H₄ perpendicular to the tpy group to reduce electrostatic forces and steric interference. Noting these two

configurations, proposed chemical structures are shown in Fig. 4c, f for the C₂H₄ prebond (flat, or tilted on-edge) and bond (vertical; standing up), respectively. Neither of the C₂H₄ prebond or C₂H₄ bond configurations appeared on samples where only CO was dosed. Additionally, pulsing experiments confirmed the presence of C₂H₄ in the prebond and bond configurations, and demonstrated pulse-induced switching between prebond and bond structures (see Supplementary Fig. 1).

## Scanning Tunnelling Spectroscopy of Fe-terpyridine with CO and C₂H₄

To confirm the formation of new chemically bonded species, we attempted STS on all 6 identified structures, see Fig. 5. Topographs and dI/dV maps showing the regions averaged over to obtain the STS in Fig. 5 are provided in Supplementary Fig. 2. The STS for bare tpy and Fe-tpy sites are shown in Fig. 5a for reference. Coordination of an Fe atom to the tpy group gives rise to an enhanced differential conductance around $-0.09$ V (below the Fermi energy) localized at the Fe site, indicating a new Fe-centred state as previously reported for a similar molecule[50,51]. STS on CO and C₂H₄ bonded structures show new pronounced tunnelling resonances at $-0.16$ and $-0.30$ V, respectively, and no longer show an identifiable feature at $-0.09$ V. This downward shift of the Fe-centred state indicates a stabilization of the Fe-centred HOMO of the Fe-tpy complex as would be expected upon formation of a new chemical bond. Further, the sharpening and increased intensity of the Fe-centred state following bonding to the gaseous reactants is likely due to an increased Fe distance above the Ag(111) surface reducing electronic interaction with the delocalized electrons of the Ag(111) and surface state. The CO prebond state was too unstable to reliably perform STS. However, STS on the C₂H₄ prebond state shows no discernable change in the Fe-centred state over the range of biases stable spectroscopy could be obtained, indicating that there is a weak or negligible electronic/bonding component to the interaction between the Fe and C₂H₄ in this structure. The absence of a bonding-induced shift of the Fe-centred state, which would indicate a significant degree of electronic hybridization, for the prebond structure further supports our proposed surface-bound intermediate, en route to a Fe-bonded species.

## Discussion

Based on the progression of observed structures with annealing, both CO and C₂H₄ appear to attach to Fe-tpy sites via a metastable

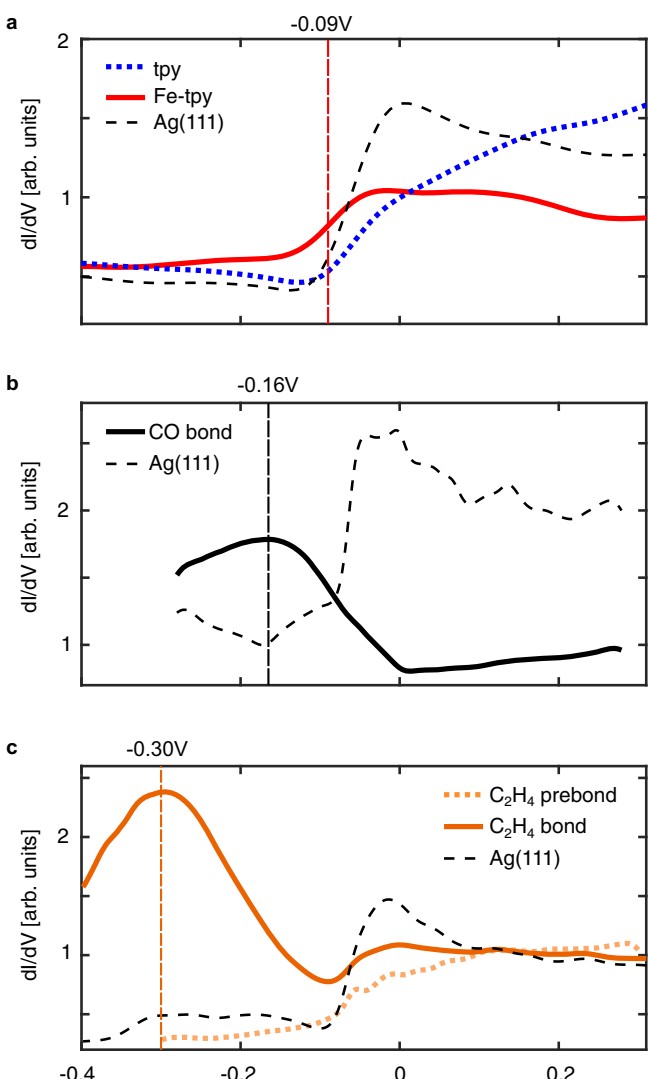

**Fig. 5 | Tunnelling spectroscopy of Fe sites.** Averaged d$I$/d$V$ comparing the bare active site signal (**a**, red solid line; position identified from comparing tpy-Fe relative to ligand) with the CO bond (**b**, black solid line), $C_2H_4$ prebond (**c**, orange dotted line), and $C_2H_4$ bond (**c**, orange solid line) tunnelling resonances. A clear shift in the Fe-centred state at $V = -0.09$ V (**a**) is observed upon CO bond formation at $V = -0.16$ V (**b**) and $C_2H_4$ bond formation at $V = -0.30$ V (**c**). The $C_2H_4$ prebond (**c**) does not exhibit deviation from the Fe-tpy resonance at V = −0.09 V. Spectra are spatially averaged from a grid spectroscopy measurement over a radius of 10 pixels (357 spectra) for the Ag(111) reference and 7 pixels (185 spectra) for all others.

intermediate. Considering a picture where the molecules first adsorb on the surface and diffuse to an active site, we propose the aforementioned "prebond" metastable intermediate where the dominant interaction remains with the Ag(111) substrate before converting to a more stable Fe-bound state. The proposed structures above were based on these two extremes of interaction: a surface-dominant interaction leading to an adsorption-like configuration, and an Fe-dominant interaction leading to a gas-phase-like configuration. CO exhibits a "standing up" adsorption configuration on most metal surfaces, including Ag(111)[61], with the carbon atom bound to the surface. The same carbon–metal bonding motif is expected with Fe leading to a "sticking out" configuration for the complex. For our surface-bound Fe-tpy sites, this requires some thermal energy to break the Ag–CO interaction and allow rotation of the molecule leading to a barrier for binding. For $C_2H_4$, the expected adsorption geometry is with the π-

system parallel to the Ag surface[60], while the expected gas-phase configuration would have the $C_2H_4$ π-system bonded to the Fe and the $C_2H_4$ orthogonal to the tpy ligand plane. Here, additional thermal energy is also needed to allow the $C_2H_4$ to first diffuse—compared to CO which diffuses at ~5K[62]—to the Fe sites so it is possible that a partial rotation already occurs before the pre-bond stage to overcome the steric interference of the H-atoms with the tpy moiety to give the observed structure. The ncAFM images appear to corroborate this picture, showing structures consistent with surface-bound CO and $C_2H_4$ adjacent to the Fe site converting to structures consistent with the expected gas-phase bonding to Fe-tpy.

To supplement this empirical picture, density-functional theory (DFT) calculations including dispersion forces[63–68] were carried out to find optimized structures for a single terpyridine-Fe moiety on a Ag(111) surface and to explore and compare the relative stability of different configurations of CO and $C_2H_4$ close to a Fe-tpy site. Due to the substantial non-covalent interactions for molecular systems such as these, dispersion interactions are essential to obtaining reliable structures. The relaxed structures were then used to compute simulated STM and ncAFM images to directly compare to the data (Fig. 6, side views of structure shown in Supplementary Fig. 5). Details of the DFT and image simulations are given in the methods. The nascent Fe-tpy structure is consistent with the full ligand and previous work[50,51]. While the presence of a metal surface complicates interpretation of charge states, the Bader charge of 0.818e⁻ (see Supplementary Table 1 for Bader analysis of all structures) found here is consistent with previous results corresponding to a $Fe^{2+}$ state found by NEXAFS for the coordinated iron with a similar tpy-based ligand[51]. The bare Fe-tpy structure, and all others except the CO-bonded structure, are found to have a triplet ground state.

Two configurations were observed for CO in the simulations: a vertical configuration consistent with a surface-bound intermediate, and a more stable horizontal configuration consistent with the expected gas-phase tpy-Fe–CO complex. The simulated STM and ncAFM images show strong agreement with the data, reproducing the change in intensity near the Fe–CO bonded site in STM, and clearly reproducing the change in line length in ncAFM imaging associated with the change in CO orientation. Differences in the experimental and modelled changes in this feature are likely due to relaxations of the surface structures ignored in the point-probe model. The vertical orientation of the CO, bonded simultaneously to the Ag and Fe site, implies a side-on interaction between the CO and Fe in an $\eta^2$ motif. While not unprecedented, as there are examples of $\eta^2$-carbonyls isolated in multimetallic clusters[69,70], it is unusual. Here, the substrate helps stabilize this structure in addition to a covalent component to the CO–Fe interaction; breaking down the binding energy for the CO vertical structure (see Supplementary Table 2) one finds of the overall 17.6 kcal/mol, 7.1 kcal/mol arises from dispersion and 10.5 kcal/mol is from the base functional. Experimentally, CO binds to the open Ag surface by 6.5 kcal/mol[71], so over 1/3, or at least 11.1 kcal/mol must arise from the interaction with the Fe-tpy site comprised of both dispersion and electronic interaction. Additionally, there is a significant stretch of the C-O bond, with a short Fe−C distance implying coordination to the Fe. To further investigate, single-point calculations of a free Fe-tpy site (no surface) with an $\eta^2$-CO with fixed coordinates showed an even stronger interaction of 28 kcal/mol, higher even than the CO vertical case (likely due to a reduction of Fe reactivity adjacent to the Ag surface), pointing to the potential for this binding motif if stabilized in some way. As such $\eta^2$-carbonyls have been studied in the context of activation of CO via bridging complexes, this provides a potentially interesting intermediate enabled by the surface metal atoms adjacent to the active Fe-tpy. As this intermediate converts to the CO-bond form, it becomes significantly more stable and has a singlet ground state due to the strong ligand field of the CO (see Supplementary Information for additional discussion).

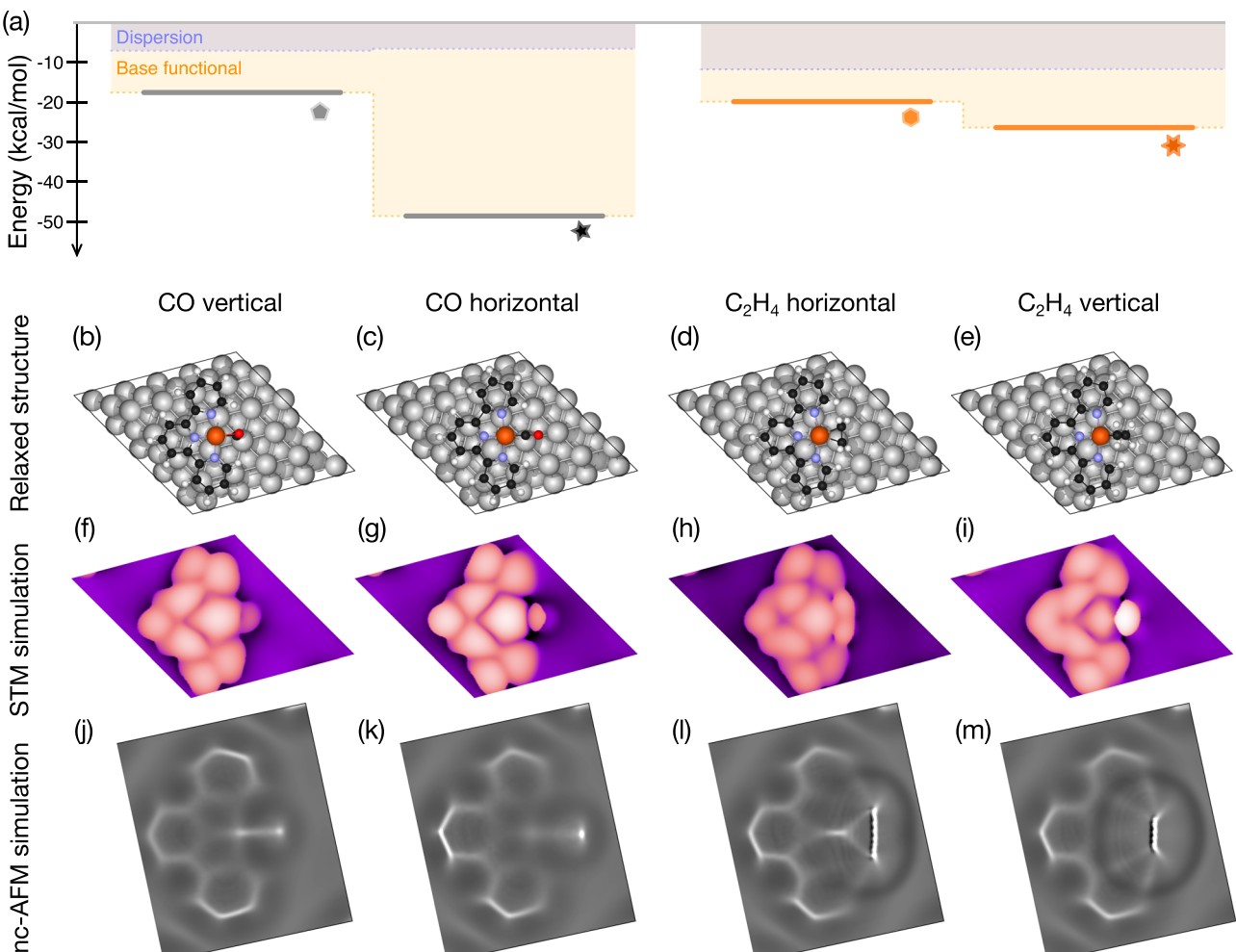

**Fig. 6 | DFT modelling of 4 most commonly observed Fe-tpy bound structures.** Dispersion corrected DFT simulation results for 4 most common stable structures showing binding energies relative to bare Fe-tpy and dispersion and base functional contributions for each (**a**) with optimized structure (**b**–**e**) and simulated STM (**f**–**i**) and Laplace filtered nc-AFM (**j**–**m**) shown below for each to compare with data. Comparing CO vertical and CO horizontal simulation results to Fig. 3, these most closely match the CO pre-bond and bond structures respectively with corresponding increase in stability as expected from observations. $C_2H_4$ horizontal and $C_2H_4$ vertical simulation results compare well with Fig. 4 $C_2H_4$ pre-bond and $C_2H_4$ bond respectively, again with increase in stability, albeit smaller consistent with experimental observations. All structures except for the CO-horizontal were found to have a triplet ground state consistent with a weak ligand-field picture.

The two most stable $C_2H_4$ structures found were a horizontal and vertical orientation of the $C_2H_4$ bound to the Fe site, which again reproduce both the STM and ncAFM imaging shown in Fig. 6; in particular, the handle-like feature in ncAFM imaging of the $C_2H_4$ prebond structure is well reproduced by the $C_2H_4$ horizontal configuration. The computed pre-bond (horizontal) structure interestingly shows a 90° rotation of the $C_2H_4$ from the expected flat-lying adsorption configuration on Ag(111), indicating that $C_2H_4$ undergoes diffusion and rotation on the surface when given thermal energy near the Fe site to form the observed "pre-bond" intermediate, and further rotation to convert to the most stable Fe-dominant interaction upon bonding to the Fe-tpy sites. Notably, similar to the CO pre-bond and bond structures, the dispersion contribution is similar for both of these structures, and the overall gain in stability arises from additional base-functional contribution indicating a stronger electronic bond with Fe upon conversion.

Stability of the four structures was experimentally probed through counting the proportion of sites in each state after increased annealing time and temperature, and through observation of scanning and voltage-pulse induced dissociation (see Supplementary Fig. 1) of the CO or $C_2H_4$ from the Fe-tpy site. The CO prebond, which frequently formed during the deposition conditions with $T = 11$ K, was readily dissociated or converted to the CO bond structure even under gentle imaging and STS conditions. In contrast, dissociation of the CO bond structure was not observed without substantial disruption to the tip and region probed. Thermally-induced conversion from CO prebond to CO bond occurred predominantly above ~30 K, with a shift to an increasing proportion of CO bond with increasing annealing time and temperature. The $C_2H_4$ prebond and $C_2H_4$ bond structures were only rarely observed prior to annealing to at least 25 K, likely due to the larger diffusion barrier and rotational requirement to interact with the Fe-site consistent with the DFT modelling. With increasing annealing time and temperature, again the population shifted from prebond to bond, indicating a small barrier to this configuration change, but a more stable bond structure, consistent with the rotation to a vertical configuration suggested by the DFT. Pulsing experiments showed that both forward and backward conversion of the $C_2H_4$ structures, as well as dissociation, was possible, indicating a weaker reactant-Fe bond than in the CO-bond structure. This leads us to conclude an overall experimental stability order from weakest to strongest of: (1) CO prebond, (2) $C_2H_4$ pre-bond, (3) $C_2H_4$ bond, (4) CO bond. This is consistent with the stability order determined from the binding energies of the relaxed Fe-tpy structures (Fig. 6a and Supplementary Table 2) found to be: −17.6 kcal/mol for CO-vertical (CO pre-bond),

−19.9 kcal/mol for $C_2H_4$-horizontal ($C_2H_4$ prebond), −26.4 kcal/mol for $C_2H_4$-vertical ($C_2H_4$ bond), and −48.6 kcal/mol for CO-horizontal (CO bond).

Through STM, STS, and ncAFM measurements of the different species observed before and after dosing of small-molecule reactants, and comparison with the 4 most stable structures calculated via DFT, we have determined that the gaseous small-molecule reactants attach to our surface-bound Fe-tpy reactive sites via metastable surface-bound intermediates. The anneal-and-quench method used to promote diffusion and reactivity allowed observation of the progression of these intermediates at individual sites that can also be combined to build an ensemble view of the reaction progression and stability of intermediates. Remarkably, we found the two extremes of an adsorption-like configuration and a gas-phase-like configuration provided a reasonable heuristic for a surface-bound intermediate progressing to a final bound state, providing a simple conceptual starting picture for these systems. Nevertheless, in cases where the balance of surface and metal-organic site interactions are similar, and where sterics play a role, as with ethylene here, the intermediate may adopt a different configuration from the surface adsorption and additional bonding mechanisms may be at work, as with the $\eta^2$-CO configuration of the CO prebond previously reported for multimetallic clusters. Likewise, while familiar conceptual models like ligand-field theory provide guidance for solution chemistry of metal-organic complexes, the asymmetric geometry and influence of the surface make this approach insufficient here. Elucidating such details requires full surface calculations and consideration of both covalent and non-covalent interactions. While interaction with the surface complicates design and implementation of surface-supported single-site catalysts, the opportunity lies in including the surface in design of reactivity to access otherwise challenging reaction pathways; a cluster perspective including surface atoms may open the door to advancing both control and complexity.

Our investigation of this self-assembled, surface-bound metal-organic reactive site demonstrates the importance of including the reactant-surface interaction, as well as the reactant-catalyst interaction, when considering anchored molecular single-site heterogeneous catalysts. By investigating dilute sites and reactants, and operating at low temperatures to slow down reaction progression, we were able to visualize the early stages of these reactions. Interestingly for this particular system, remarkably low-temperature reactivity (below 40 K) was observed. While surface-anchored metal-organic catalysts offer exciting prospects for molecular design and control of reactivity, our findings emphasize the importance of the substrate when comparing reaction pathways in surface-anchored metal-organic catalysts to those of their solution-based counterparts.

## Methods
### Active site preparation
The Ag(111) surface was cleaned by $Ar^+$ sputtering, and annealing at $T = 690$ K. Terpyridine-phenyl-terpyridine (TPT; SigmaAldrich, 96%) molecules were sublimed at $T = 490$ K onto the room temperature surface. Typical deposition times of 30 s resulted in an average surface coverage of 0.04 TPT/$nm^2$. Pressure during TPT deposition was <$2.0 \times 10^{-10}$ mbar. The deposition time for Fe was selected to achieve a -1Fe:1TPT ratio based on observed TPT coverage and an Fe deposition rate of -$9.0 \times 10^{-4}$ Fe atoms $s^{-1} nm^{-2}$ estimated from STM imaging for an e-beam evaporator flux of $I_{flux} = 2.3$ nA. Fe was deposited on the Ag(111)/TPT sample held at room temperature in short bursts−alternating $t = 5$ s shutter open with $t = 15$ s shutter closed−to reduce Fe cluster formation. Samples were returned to the 4.2 K STM head immediately following Fe deposition to minimize Fe-TPT chain formation. The pressure during Fe deposition was <$5.0 \times 10^{-10}$ mbar. For all imaging and characterization experiments, the chamber pressure was ≤$7.0 \times 10^{-10}$ mbar.

### Prebond and bond formation
CO and $C_2H_4$ were introduced via a leak valve connected to the low-temperature imaging chamber after active site preparation. Either CO or $C_2H_4$ was leaked into the chamber at pressures ranging from $2.0 \times 10^{-10}$ mbar to $3.0 \times 10^{-10}$ mbar for up to 3 min with sample temperatures reaching up to $T = 11.6$ K during dosing. Higher pressure, shorter dosing was intended to minimize the sample temperature increase from the base temperature of $T = 4.2$ K; however, there were no notable differences in the resulting structures. Samples containing active sites and gaseous CO and/or $C_2H_4$ adsorbates were annealed from $T = 15$ K to $T = 40$ K incrementally for 5 min intervals with $\Delta T = 5$ K via an in-situ heater on the STM stage.

### STM/STS/ncAFM measurements
STM, STS, and ncAFM data were acquired at 4.2 K in an Omicron LT-SPM. All STS measurements were performed with a Pt/Ir tip, likely terminated with Ag. To spatially resolve the local density of states, $I(V)$ spectra were acquired at each $(x, y)$ pixel. Spatial regions of the $(x, y)$ grid that displayed uniform tunnelling resonances were averaged, low-pass filtered in V, numerically differentiated, and rescaled by dividing by the setpoint parameters: $dI/dV/(I_s/V_s)$. NcAFM measurements were performed with CO-terminated W tips on a homebuilt qPlus sensor. Amplitude calibration was performed following Simon et al.[72], details in SI. To terminate the tip with CO, $df(z)$ sweeps incrementing 1 Å from an initial $\Delta z = 5$ Å were performed over a single CO molecule until it was transferred to the tip. STM topographies were acquired with both types of tips as indicated in the captions. Images were processed in WSxM[73].

### DFT modelling
All calculations were performed using the projector augmented-wave (PAW) method[64], with the B86bPBE density functional[63,65] and the XDM dispersion correction[67] as implemented in Quantum ESPRESSO[68], version 5.1.1. The Ag(111) surface was modelled as a 4-layer, $4 \times 4$ supercell, with lattice constants matching our previous work[66]. The adsorbed complex was taken as a single Fe atom coordinated to a terpyridine molecule. Both singlet and triplet states of this Fe-tpy complex, with and without CO or $C_2H_4$, were considered by applying an initial spin bias to the Fe atom. The triplet was consistently more stable, except for the CO-bonded case.

The atomic positions of the Fe-tpy complex were first relaxed using only the Gamma point, with planewave cutoffs of 50 and 400 Ry for the kinetic energy and charge density, respectively, with the Ag surface held fixed. The k-point mesh was then increased to $2 \times 2 \times 1$ and the top two layers of the Ag (111) surface were also allowed to relax. Finally, the single-point energy was evaluated with the larger k-point mesh and higher planewave cutoffs of 60 and 600 Ry. Starting from the optimized Fe-tpy structure, additional calculations were performed with CO and $C_2H_4$ placed in a number of candidate prebonding and bonding configurations using the same computational protocol. These either collapsed to the structures shown in Fig. 6, or were considerably higher in energy and not considered further. Supercell calculations at the Gamma point for isolated CO, $C_2H_4$, and the triplet Fe-tpy molecule were also performed with the same cutoffs to obtain the binding energies.

### Image simulation
Constant-current STM plots were generated from post-self-consistent calculations, including 1.2 times the number of occupied bands, within the Tersoff–Hamann approximation[74] using the Critic2 programme[75]. A sample bias of −0.02 eV was selected to match experiment.

NcAFM plots were simulated using the method developed by Hapala et al.[76]. The metal-oxygen bond used to model the CO-functionalized tip was approximated by a pairwise Lennard-Jones (LJ) potential, with OPLS force field parameters[77] and a lateral and radial

stiffness of 0.24 and 20.0 N/m, respectively[78]. The tip is relaxed 0.7 nm above the sample surface until the net force on the probe is less than $10^{-6}$ eV/Å. The electrostatic component of the force field is calculated from the surface Hartree potential[79], while van der Waals and Pauli repulsion contributions are approximated by the LJ potential[76]. The resulting images were Laplace-filtered to compare with experimental ncAFM images.

## Data availability

The data used in this study have been deposited in the Open Science Framework repository with associated https://doi.org/10.17605/OSF.IO/2TKE6 and direct link here: https://osf.io/2tke6/.

## Code availability

The critic2 code is distributed under the GNU General Public License and is available from https://github.com/aoterodelaroza/critic2 with https://doi.org/10.5281/zenodo.7268767.

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

## Acknowledgements

Work done at UBC was supported by NSERC Discovery Grants Programme (RGPIN-2018-04271), the ACS Petroleum Research Fund (55955-ND5), Canadian Foundation for Innovation, Canada Research Chairs programme (SB), the Stewart Blusson Quantum Matter Institute, the Canada First Research Excellence Fund Quantum Materials and Future Technologies Programme, and the Swedish Research Council (VR) 2016-06719 (EM). Computational work was supported by NSERC (RGPIN-2016-05795), and through computational resources provided by Compute Canada. We would also like to thank Brandon Stuart for designing and constructing the gas dosing system.

## Author contributions

S.B. and M.D. designed the experiments and wrote the manuscript. M.D. performed the experimental data collection, analysis, and majority of writing. G.T. performed the ncAFM simulations, and assisted with data collection and instrument maintenance. E.M. and G.N. assisted with data collection, instrument maintenance, and guidance with experimental

design, planning, and interpretation. A.J.A.P. performed the DFT simulations and computed the simulated STM images. E.J. provided supervision and guidance for the theoretical components, while S.B. provided supervision and guidance for the experimental components. All authors contributed to the discussion and interpretation of the data and theoretical results, and contributed to the final manuscript.

## Competing interests

The authors declare no competing interests.
