## [Peer Review File · Nature Communications]

Small molecule binding to surface-supported single-site transition-metal reaction centresEditorial Note: This manuscript has been previously reviewed at another journal that is not operating a transparent peer review scheme. This document only contains reviewer comments and rebuttal letters for versions considered at Nature Communications.

REVIEWERS' COMMENTS

Reviewer #1 (Remarks to the Author):

The authors have taken my comments on their earlier submission into account and addressed them extensively and convincingly. I can now support publication in Nature Communications after the following additional comments have been addressed:

- In their answer to my comment on the oxidation state of the Fe-tpy complex, the authors write that they have added a piece of clarifying text to the manuscript (“while the node...”). This piece of text is however absent in the manuscript, and I strongly recommend its addition to avoid confusion about the redox state of the initial iron complexes.

- In their answer on my comment about the binding motifs of the ethylene fragment to the iron complexes, the authors have added side-on views of the models to the SI (in Fig S4). As I found these very insightful to understanding the discussion, I suggest the authors to include these side-on views in Fig. 6 of the main manuscript instead of in the SI.

Reviewer #2 (Remarks to the Author):

Dear Editor

The authors have significantly improved their manuscript and addressed previous concerns. Specifically, the AFM simulation of a vertical C₂H₂ adsorption site (Figure 6m) was changed considerably and now strongly supports the experimental interpretation. Since such experiment-simulation correspondence (for both bond and prebond) is now striking, the authors are encouraged to further discuss the ‘matching’ AFM distortions/features in the main text. For instance, are such matching features geometrical or electronical? Though the authors argue that care should be taken when ‘extracting too much from the probe-particle model simulations as they rely on relatively simplistic force field approximations’, this Reviewer considers that discussing structural data in detail within the limits of the approximation, is fundamental to advance knowledge of ‘surface-supported reaction centres’.

In this regard, it can be additionally recommended to include the previous (‘non-matching’) AFM simulation of a vertical C₂H₂ adsorption site in e.g. Supplementary Note 10 or as a Supplementary Note 11, and discussed appropriately.

In view of the strengthened conclusions, this reviewer considers that this work is ready for publication. The abovementioned remarks are simply suggestions which might be very much welcomed by the community.

Minor comments:

o Does Fig. 6 show AFM simulations after Laplace transform? If so, shouldn't this be clear in the captions and the data without Laplace transform included in Supplementary Note 10, as Supplementary Note 11, etc. instead of in the data repository?

o This Reviewer found the authors' reply letter statements related to the multiplicity of the surface-supported reactions centers, e.g. ‘Fully filling the two lowest-energy orbitals and half-filling the next two orbitals explains the observed triplet spin state’ worthy of further comment. The authors might consider discussing such arguments in the main text.

Reviewer #3 (Remarks to the Author):

The authors addressed the comments by the referee, and hence I can recommend the manuscript for publication. First, however, I would recommend taking the following points into account:

- In the Laplace filtered images, the min and max are confusing as the filtered image does not directly represent df .
- In the CO example (Fig. 3), the extension of the longer feature is measured to change from 2.6A to 4.5A. The authors should report how well this is reproduced in the calculated image, where this change seems smaller. And eventually, give some explanation for the deviation.
- The dI/dV curves in Fig. 5 are averaged. It should be mentioned how many curves were averaged.

We again thank the reviewers for their time and thoughtful comments. We have adjusted the manuscript according to these suggestions where applicable and have recorded our responses below.

Reviewer #1 (Remarks to the Author):

The authors have taken my comments on their earlier submission into account and addressed them extensively and convincingly. I can now support publication in Nature Communications after the following additional comments have been addressed:

- In their answer to my comment on the oxidation state of the Fe-tpy complex, the authors write that they have added a piece of clarifying text to the manuscript (“while the node...”). This piece of text is however absent in the manuscript, and I strongly recommend its addition to avoid confusion about the redox state of the initial iron complexes.

The quote in the previous referee response beginning with “while the node” is taken from reference [51] Krull et al. Nature Commun. 3211 (2018), which we cite in the two statements below that were added to our manuscript:

Text was added on Pg 3:

“Prepared entirely in ultrahigh vacuum (UHV), the iron is expected to have an incomplete coordination sphere, and a nominal charge state of 2+ based on previous results with a similar terpyridine ligand⁵¹, differing from that of pure Fe atoms¹⁷ or clusters on the surface. The electronic structure and charge state of this complex are addressed further in the discussion.”

And in the discussion on Pg. 10-11:

The nascent Fe-tpy structure is consistent with the full ligand and previous work^{50,51}. While the presence of a metal surface complicates interpretation of charge states, the Bader charge of 0.818e- (see SI table S1 for Bader analysis of all structures) found here is consistent with previous results corresponding to a Fe²⁺ state found by NEXAFS for the coordinated iron with a similar tpy-based ligand⁵¹. The bare Fe-tpy structure, and all others except the CO-bonded structure, are found to have a triplet ground state.

- In their answer on my comment about the binding motifs of the ethylene fragment to the iron complexes, the authors have added side-on views of the models to the SI (in Fig S4). As I found these very insightful to understanding the discussion, I suggest the authors to include these side-on views in Fig. 6 of the main manuscript instead of in the SI.

As Fig. 6 is already very crowded, and the vertical orientation is preferable for all other structures we have decided to leave the side-on figures in supplemental with the description in the text to accompany the schematics and structure plots in Fig. 6.

Reviewer #2 (Remarks to the Author):

Dear Editor

The authors have significantly improved their manuscript and addressed previous concerns. Specifically, the AFM simulation of a vertical C₂H₂ adsorption site (Figure 6m) was changed considerably and now strongly supports the experimental interpretation. Since such experiment-simulation correspondence (for both bond and prebond) is now striking, the authors are encouraged to further discuss the ‘matching’ AFM distortions/features in the main text. For instance, are such matching features geometrical or electronic? Though the authors argue that care should be taken when ‘extracting too much from the probe-particle model simulations as they rely on relatively simplistic force field approximations’, this Reviewer considers that discussing structural data in detail within the limits of the approximation, is fundamental to advance knowledge of ‘surface-supported reaction centres’.

In this regard, it can be additionally recommended to include the previous (‘non-matching’) AFM simulation of a vertical C₂H₂ adsorption site in e.g. Supplementary Note 10 or as a Supplementary Note 11, and discussed appropriately.

While we agree that such data showing discrepancies should normally be shown, we were unable to reproduce the original modeling results within reasonable parameters. Since the parameter space that most closely matches the experimental conditions consistently produces features similar to those now shown, we believe it is confusing and potentially misleading to include the old results as part of the manuscript or supplemental. They are provided here for the review record.

Review figure (left): previous version of Figure 6, updated with modeling more accurately representing experimental parameters and additional information and formatting to more clearly convey the energetics of the different structures.

In view of the strengthened conclusions, this reviewer considers that this work is ready for publication. The abovementioned remarks are simply suggestions which might be very much welcomed by the community.

Minor comments:

o Does Fig. 6 show AFM simulations after Laplace transform? If so, shouldn't this be clear in the captions and the data without Laplace transform included in Supplementary Note 10, as Supplementary Note 11, etc. instead of in the data repository?

The supplementary figure 6 contains the unfiltered simulation. This has been clarified in the caption captions in both the manuscript (figure 6; with Laplace filtering) and in the Supplementary information (supplementary figure 6; without Laplace filtering). This was done intentionally to provide the comparison with the Laplace filtered data in the manuscript but also provide the unfiltered simulation.

o This Reviewer found the authors' reply letter statements related to the multiplicity of the surface-supported reactions centers, e.g. 'Fully filling the two lowest-energy orbitals and half-filling the next two orbitals explains the observed triplet spin state' worthy of further comment. The authors might consider discussing such arguments in the main text.

This was considered, however as this orbital assignment is not clearly well-defined in the presence of a metal substrate strongly interacting with the adsorbed structure, we felt it best to discuss only the spin state in the manuscript and leave the MO-like diagram derived from the DFT to the SI only. This may be overly cautious, but we did not feel it was a sufficiently accurate representation to feature in the main text. Nevertheless, we are pleased to hear the reviewer connected with this representation and hope it is helpful to others.

Reviewer #3 (Remarks to the Author):

The authors addressed the comments by the referee, and hence I can recommend the manuscript for publication. First, however, I would recommend taking the following points into account:

-In the Laplace filtered images, the min and max are confusing as the filtered image does not directly represent df.

Understood, however, it is our preference (and in accordance with the journal guidelines) to ensure the reader can clearly associate the colour scale with the intensity as it is also possible to have plotted this with different or inverted scales. We have left the scales as "min" and "max" as they do not directly correspond to df values.

-In the CO example (Fig. 3), the extension of the longer feature is measured to change from 2.6Å to 4.5Å. The authors should report how well this is reproduced in the calculated image, where this change seems smaller. And eventually, give some explanation for the deviation.

Indeed there is a deviation. As we cannot know the exact height above the molecule or the stiffness of the CO, there is some guesswork in setting the parameters of the point-probe model. Notably our original simulations with what we believe to be less realistic parameters showed a longer protrusion likely due to an overly small stiffness of the tip-CO. It seems this may then be a result of the relaxation mechanisms of the tip-CO interacting with surface-bound structures, including possibly surface structure relaxation. Note that in the point-probe model, the surface structure is NOT relaxed. A sentence has been added to the manuscript to address this point: "Differences in the experimental and modelled changes in this feature are likely due to relaxations of the surface structures ignored in the point-probe model."

-The dI/dV curves in Fig. 5 are averaged. It should be mentioned how many curves were averaged.

All STS over molecular features (tpy, Fe-tpy, CO-bond, C2H4-prebond and C2H4-bond) are averaged spatially from grid spectroscopy data (i.e. $dI/dV(x,y,V)$) over a circle of radius 7 pixels corresponding to 185 points; areas are shown in SI for each structure. Reference Ag spectra are averaged over a circle of radius 10 pixels corresponding to 357 points from the same image (i.e. corresponding to the same tip condition).